# Channeled Polarimetry for Magnetic Field/Current Detection

**DOI:** 10.3390/s25020466

**Published:** 2025-01-15

**Authors:** Georgi Dyankov, Petar Kolev, Tinko A. Eftimov, Evdokiya O. Hikova, Hristo Kisov

**Affiliations:** 1Institute of Optical Materials and Technologies, Bulgarian Academy of Sciences, 109, Acad. G. Bonchev Str., 1113 Sofia, Bulgaria; gdyankov@iomt.bas.bg (G.D.); ebelina@iomt.bas.bg (E.O.H.); hristokisov@iomt.bas.bg (H.K.); 2Central Laboratory for Applied Physics, Bulgarian Academy of Sciences, 61 Sanct Petersburg Blvd, 4000 Plovdiv, Bulgaria; tinko.eftimov@uqo.ca; 3Centre de Recherche en Photonique, Université du Québec en Outaouais, 101 rue Saint-Jean-Bosco, Gatineau, QC J8X 3X7, Canada

**Keywords:** channeled polarimetry, magnetic field sensor, current sensor, spectral interrogation

## Abstract

Magneto-optical magnetic field/current sensors are based on the Faraday effect, which involves changing the polarized state of light. Polarimetric methods are therefore used for measuring polarization characteristics. Channeled polarimetry allows polarization information to be obtained from the analysis of the spectral domain. Although this allows the characterization of Faraday materials, the method has not yet been used for detection in magneto-optical sensors. This paper reports experimental results for magnetic field/current detection using the channeled polarimetry method. It is shown that in contrast to other methods, this method allows the detection of the phase shift caused by Faraday rotation alone, making the detection independent of temperature. Although an increase in measurement accuracy is required for practical applications by refining the data processing, the experimental results obtained show that this method offers a new approach to improving the performance of magneto-optical current sensors.

## 1. Introduction

In recent decades, the application of magneto-optical current sensors has become an important branch of magnetic field/current detection. In high-voltage applications in a power electric system, the advantages of optical sensors over traditional magnetic systems include improved safety, smaller size, better and wider frequency band, and broader dynamic range. Their inherent immunity to electromagnetic interference allows them to be used in high-voltage electrical power systems. A very important feature of magneto-optical current sensors is the absence of saturation effect (hysteresis), which allows them to be used as highly efficient protection transformers.

In magneto-optical sensors, magnetic field measurement is based on the linear magneto-optical effect (Faraday effect), where the rotation of the polarization angle of light propagating in the magneto-optical material is induced by an external magnetic field.

There are various interrogation techniques that can be used to detect the angle of rotation of polarization. Two basic schemes are used to measure the azimuth of the magnetic field-induced (Faraday) polarization rotation: polarimetric and interferometric.

The simplest polarimetric scheme consists of a polarizer and an analyzer with the magneto-optical material between them, as shown in Figure 1 in [1]. Various signal analysis techniques are used to measure the Faraday rotation angle. In ref. [1], a wavelength-dependent interrogation method was explored, ensuring the conversion of the Faraday rotation into a spectral response.

Interferometric detection schemes measure the Faraday rotation angle in terms of the phase difference between the two orthogonal circular modes (left and right circular polarization). The Sagnac interferometer is widely used for current detection in fiber-optic magnetic field sensors [2].

Magneto-optic materials are crystals or doped glasses that have a remarkable Faraday magneto-optic effect (high Verdet constant) and a low absorption coefficient. The Bi_12_SiO_20_ (BSO) crystal was established as a promising magneto-optic material in the early 1990s. The first study of the properties of a BSO crystal as a sensing element in a current sensor was carried out in [3].

In ref. [4], it was found that BSO has high optical activity due to the low symmetry of the polyhedron chiral complex. Moreover, the optical activity in the [*100*] and [*110*] directions is of the same magnitude, which is convenient for practical applications since no crystal orientation is required. As for the doping materials, they were shown to increase the optical activity near the absorption shoulder [5] and from a practical point of view some of them are suitable for photorefractive applications. Addressing the influence of dopants on the Verdet coefficient, Fe with a suitable concentration was found to increase the coefficient; however, the BSO:Fe crystal possesses nonlinear dependence of the Faraday rotation on the magnetic field [6]. A similar conclusion has been drawn recently with regard to BSO:Ni [7]. This makes undoped BSO crystals a very good magneto-optical material. They have a high Verdet coefficient, no double refraction, and exhibit low optical losses.

BSOs have high optical activity, which has been shown to be the main cause of erroneous results in magnetic field detection using BSO crystals. On the other hand, this quality can be used to improve accuracy. The spectral interrogation studied in our papers [1,8] is based on the periodic modulation of the broadband spectrum of the source, mainly caused by intrinsic optical activity. In ref. [9], crystals with opposite optical activities were used and an accuracy of 0.3% was achieved in field tests. In ref. [10], the optical activity of a (BGO) crystal was used to compensate for temperature drift. For this reason, the presence of optical activity can be considered to be an advantage.

The ideal BSO crystal has no birefringence. However, the latter can be caused by internal stresses, dislocations, or stresses during mounting in the holder. The value of birefringence is therefore a characteristic specific to each sensing element. Birefringence converts linear polarization into elliptical polarization, resulting in intensity variations with temperature. However, as shown in [11], birefringence, as well as optical activity, allow temperature compensation methods to be applied.

As early as the inception of the development process, the issue of temperature stability in magneto-optical sensors, particularly those based on BSO, has been identified [3]. It has been demonstrated that the temperature drift is exclusively dictated by the temperature-dependent Verdet coefficient. Subsequently, it has been established [12] that, under specific polarizer and analyzer orientations that align with the crystal length, the temperature dependencies of the Verdet coefficient and the intrinsic optical activity can offset each other. The analysis of magneto-optical sensors in terms of their temperature stability demonstrates that temperature compensation should be performed with an accuracy of 0.1% and 1% when utilizing them as metering and protective transformers, respectively [13]. However, the realization of these requirements in actual operating conditions is challenging, given the limited widespread application of optical sensors to date.

In this study, the channeled polarimetry method [14,15] was employed to explore avenues for enhancing the performance of magneto-optical sensors. The experimental design involved the utilization of the same sensing element as in our previous studies [1,8], ensuring a direct comparison and facilitating a comprehensive interpretation of the results. We demonstrated a direct detection of a phase shift caused solely by the Faraday rotation. This is very different from the polarimetric detection methods known to date, in which the Faraday rotation is converted into changes in the intensity or in the spectrum. The channeled polarimetry method is considered to be polarimetric, but it also provides information about the phase shift, as do interferometric methods. The channeled polarimetry makes it possible to determine the operational points of the output state of polarization (SOP). These were found to depend on the birefringence of the BSO crystal induced, probably, during its machining. This in turn allowed these SOPs to be made independent of temperature by tuning the input polarization. The final result is a temperature-stabilized detector of magnetic field/electric current.

## 2. Principle of Operation and Experimental Arrangement

### 2.1. Channeled Polarimetry

The method of spectral modulation [14], also referred to as channeled polarimetry, is based on polarization interference to produce amplitude modulation in the spectral or spatial domain. Since amplitude modulation causes periodic modulation of the signal, the information of interest is distributed among channels of different frequencies. Channeled polarimetry uses the dispersion of birefringence in thick waveplates to convert the polarization state of the input light into intensity modulations across the light spectrum. The modulated spectrum is detected with a standard dispersion spectrometer to allow simultaneous measurement of the spectrally distributed Stokes vector in a single measurement. In effect, polarization interference occurs, which allows the phase of each component in the complex amplitude to be maintained before intensity is detected. Thus, the amplitude and phase of the Stokes components are encoded as the amplitude and phase of the carrier frequencies, allowing the Stokes components to be recovered in magnitude and sign [16]. It is important to note that the phase terms can be determined by pre-calibration with light of a known polarization state.

One of the main advantages of channeled polarimeters over standard polarimetric techniques is that they are compact and have no moving parts. The experimental implementation is shown in Figure 1, and it is similar to that used in our previous studies [1,8].

However, in the present study, the main difference was in the placement of the BSO crystal between a polarizer and a spectral polarimeter in a standard configuration [14]: two thick birefringent retarders, R1 and R2, with fast axes orientations at 0° and 45°, respectively, and an analyzer A (a calcite Glan–Thompson polarizer), with its transmission axis defining the 0° reference frame.

The side view of the sensing element—the BSO crystal mounted in the holder—is shown in Figure 2 and illustrates the description of the sensing element given in Section 2.2. We would like to draw attention to the different configuration used in channeled polarimetry, namely the cylindrical element in which the retarders and the polarizer are mounted, forming the spectral polarimeter.

By means of Mueller calculations, it has been shown [14] that the spectrum of light passing through the channeled polarimeter is cosinusoidally modulated and contains information about all Stokes parameters. Indeed, such a spectrum was observed experimentally, as shown in Figure 3a. It was subjected to a Fourier inversion which resulted in the autocorrelation function *C*(*h*), shown in Figure 3b. All channels shown in Figure 3b were satisfactorily separated from one another over the optical path difference (OPD) axis h, since the carriers (*L*_2_ − *L*_1_), *L*_2_, and (*L*_2_ + *L*_1_) were suitably chosen. The optical path difference is the product of the retarder’s thickness and the birefringence of the material, taking into account the birefringence dispersion. That is why channel separation is more efficient in thick retarders. However, their thickness must not create spectral modulation that is unresolvable by the spectrometer. Therefore, the thicknesses of the quartz retarders were designed in a ratio of *d*_2_:*d*_1_ = 3:1 to match the resolution of the spectrometer (0.5 nm), according to the procedure described in [17].

A channel containing the S_0_ Stokes component lies centered at zero frequency. Channels for S_1_ and the complex combinations S_2_ ± iS_3_ are located at carrier frequencies *L*_2_ and *L*_2_ ± *L*_1_, respectively. They can be separated using a window filter, and by means of Fourier transform we can find the complete Stokes vector at the input of the spectral polarimeter: S_in_(λ) = [S_0_(λ), S_1_(λ), S_2_(λ), S_3_(λ)]^T^ [14].

### 2.2. Spectrally Interrogated Polarimetry

In the present study, polarimetry with spectral interrogation was used to identify patterns that had not been the focus of our previous studies but that supported the hypotheses of the present study.

The experimental set-up is shown in Figure 4 and was identical to that utilized in our previous studies [1,8]. The BSO crystal (4 × 25) mm was placed in an aluminum holder. Good thermo-contact was provided between the holder and the thermoelectric coolers, (TEC) producing temperature variation in the crystal from −20 °C to 50 °C, controlled with an accuracy of ±1 °C. A polarizer was placed in front of the crystal through which the collimated light passed, supplied by a quartz polymer optical fiber with a core diameter of 600 µm. The light source was a white halogen lamp (Ocean Optics HL-2000, Ocean Optics, Orlando, FL, USA). The light passing through the crystal and analyzer was focused by a collimator into a similar optical fiber connected to a spectrometer (Avantes AVASPEC-ULS2048CL-EVO, Avantes, Apeldoorn, The Netherlands) with 0.5 nm resolution and 400 to 900 nm range. The holder-mounted BSO crystal and the TEC were placed in the center of a solenoid which produced a homogeneous magnetic field of up to 700 G along the crystal axis. The power supply provided a current *I* in the range of −24 A to +24 A (±0.1 A) to feed the coil, as shown in Figure 1 and Figure 4.

The information, provided by spectrally interrogated polarimetry, is encoded in the periodic modulation of the broadband spectrum of the source caused by the dispersions of the optical activity and the Verdet constant of the BSO crystal. The intensity of the periodic modulation is given by the expression (1):(1)$$I=\frac{1}{4}+\{1+\cos[2(\Phi +Ɵ-\alpha )]\}$$

In (1) θ and α are the polarizer and analyzer orientations and Φ is the accumulated phase along the circularly birefringent BSO crystal (2) whose length is *L*, and Δ*β* is the propagation constant difference between the left and the right circularly polarized waves along the crystal (3):(2)Φ=ΔβL(3)$$\Delta \beta =\Delta \beta_{L}-\Delta \beta_{R}$$

The measurement was taken in the absence of magnetic field (*I* = 0) and the temperature varied in the range T = −20 °C and +50 °C. For each temperature, the polarimetric response was measured. Since the difference Δ*β* depends on both wavelength and temperature [1], its changes are manifested as a wavelength shift in the polarimetric response, as shown in Figure 5.

The purpose for these measurements is commented on below in Section 3.3.

### 2.3. Measurement Procedure

Using the set-up shown in Figure 1, we performed a series of measurements by changing the temperature T and the current *I* through the solenoid, and by measuring the channeled spectrum, we derived the Stokes parameters.

The sequence of measurements was as follows:(i)At a fixed temperature T between −20 °C and 50 °C, the current was changed in the range −24 A to 24 A. For each value of the current, the spectrum was measured, as modulated by the polarization interference (Figure 3a).(ii)After scanning the currents, a new value of temperature was chosen and step (i) was repeated.(iii)Stokes’ normalized values S_1_(λ,I)/S_0_(λ,I), S_2_(λ,I)/S_0_(λ,I), and S_3_(λ,I)/S_0_(λ,I) were extracted from stored spectra.

## 3. Results and Discussion

### 3.1. Stokes Parameters’ Determination

Figure 6 shows the spectral dependence of the normalized S_2_ and S_3_ Stokes parameters for various currents at a fixed temperature of 25 °C.

The following analysis can be made on the basis of the results obtained. The spectral dependencies of S_2_ and S_3_ showed well-expressed, specific (locus) points at certain wavelengths (labeled) where the polarization state is approximately the same for all values of the current/magnetic field. In the spectral dependence of S_1_, no locus points were observed. Instead of those detected at S_2_ and S_3_, the polarization state at varying currents and equal wavelengths exhibited proximity but not complete congruence. Given that S_1_ characterizes the orthogonal linearly polarized modes, the observed phenomena can be attributed to their distinct propagation constants. This, in turn, suggests the presence of birefringence within the crystal. This hypothesis can be substantiated through the analysis of the spectral dependence of S_3_.

The normalized Stokes S_3_ parameter provides a measure of the polarization state, as it shows the difference in intensity between the right and left circularly polarized components. The normalized Stokes parameter S_3_ varies from −1 to +1 at equal intensities of the left and right circular polarization at ΔΦ = ±90°, where ΔΦ is the phase difference between them. At (ΔΦ = 0°, 180°), S_3_ = 0 represents linear polarization. It is surprising that S_3_ is not zero, given that the input polarization is linear and that the BSO crystal is supposed to be not birefringent, being an ideal circular retarder. The observed S_3_ ≠ 0 can only be explained by the presence of birefringence in the crystal due to residual stresses, induced during the machining and mounting of the crystal in the holder. To illustrate this, we discuss ellipticity and its spectral dependence.

### 3.2. Ellipticity and Faraday Phase Shift Determination

In terms of the Poincare sphere, any possible state of polarization (SOP) can be represented by a point with two spherical angular coordinates (2η, 2ε), namely longitude (i.e., azimuth angle) and latitude (i.e., ellipticity). As has been pointed out in [18], the optical rotator can produce any outgoing SOP that has the same initial latitude, i.e., ε = 0 for input linear polarization. Any polarization state with ε ≠ 0 can be produced by adding the linear retarder effect to the rotator effect. This ideology has been later applied to the analysis of a magneto-optical current sensor [11].

In contrast to [11,18], the phase difference introduced by the two elements in our sensor was not only dependent on their orientation, but also on the dispersion due to the polychromatic light source. This explains the complicated nature of ellipticity, shown in Figure 7 and obtained from the normalized S_3_:(4)$$ɛ=\frac{Arc\;\sin{(s}_{3})}{2}$$

The state of polarization was elliptical and the phase difference between the left and right polarization was introduced by both the optical linear retarder and the rotator. The BSO crystal in our experiment was optically equivalent to an optical system consisting of two elements: a linear retarder and a rotator.

The phase difference introduced by the optical rotator comprises the intrinsic optical rotatory power *ρ* (deg/mm) of BSO and the additional rotatory power *ρ*_F_ = V_B_B caused by the Faraday effect, magnified by the crystal length *L* (V_B_ and B are Verdet constant and magnetic field, respectively). Both the intrinsic optical rotatory power *ρ* and the Verdet constant *V*_B_ depend on the wavelength and temperature. The integral phase difference *Δ*Φ can be expressed as:(5)$$\Delta \Phi \left(\lambda ,T\right)=({\Delta \Phi }_{LR}\left(\lambda ,T\right)+\varrho (\lambda ,T)\pm V_{B}(\lambda ,T)B)L$$

In expression (5) ΔΦ_LR_(λ,T) is introduced by the linear retarder; the last two terms of the equation are the phase difference introduced by the optical rotator. The sign depends on the direction of the magnetic field: in the direction of light propagation or vice versa.

It can be assumed that the introduced phase difference between two consecutive locus points is 90 degrees. This can be demonstrated by comparison with the results of spectrally interrogated measurements. We have shown that by means of the spectral interrogation technique [1], we can determine the spectral position of the minima/maxima of the polarimetric response by changing the angle of the analyzer. In the present experiment, the setting was made at *I* = 0 A (zero magnetic field) and T = 25 °C, so that the maximum of the polarimetric response was at 588 nm, as shown in Figure 8 (the black curve). Since the adjustment was performed at *I_0_* = 0 A, the change in ellipticity due to the current (magnetic field) must be considered relative to the ellipticity at *I_0_* = 0 A. The spectral dependencies expressed as (6) are shown by the colored curves in Figure 8.(6)$$\Delta {ɛ}_{n}\left(I_{n},\lambda \right)=ɛ\left(I_{n},\lambda \right)-ɛ\left(I_{0},\lambda \right)$$

The primary conclusion derived from the correlation depicted in Figure 8 is that the spectral variation in ellipticity is attributable to the phase shift between the orthogonal circularly polarized modes. Furthermore, it can be determined that the phase shift is not a consequence of the linear retarder and the intrinsic optical activity, as the term $ɛ\left(I_{0},\lambda \right)$ in Equation (6) accounts for their contribution. Consequently, it can be concluded that the Faraday rotator produces a phase shift in ε = const, the plane of Poincaré sphere, manifested as a spectral dependence of ellipticity. This was observed in our experiment and corresponds to the dependence shown in Figure 8 [17]. Indeed, the magnitude of the spectral dependence of ellipticity is the phase shift caused solely by the Faraday rotation *ρ*_F_.(7)$$\Delta {ɛ}_{n}\left(I_{n},\lambda \right)=\pm V_{B}(\lambda ,T)B_{n}L$$

The magnetic field induced by the current can be determined from the spectrally manifested Faraday phase shift. Figure 9 shows the dependence of the phase shift on the current/magnetic field for fixed wavelengths (dot curves). The Verdet constant for the corresponding wavelength is determined from [4,8]. The theoretical prediction according to (7) is shown by the solid lines and it is in good agreement with the experiment.

The observed correlation between the Δε_n_ zero-crossing points and the minima/maxima of the spectral polarimetric response deserves comments.

As mentioned, at these points, the spectral interrogation technique detects linearly polarized light with a plane of polarization consistently rotated by 90 degrees (from minima to maxima and vice versa) as a consequence of the phase difference, mainly caused by the dispersion of the intrinsic optical rotation ρ(λ) (at a constant temperature).(8)$$\Delta \Phi_{\rho }\left(\lambda ,T\right)=\rho (\lambda ,T)L$$

The contribution of the Faraday rotation V.B will be less than 1% of the intrinsic optical activity *ρ*, as estimated in [1], for the magnetic fields used in the experiment. This manifests itself as a spectral shift Δε_n_ of the zero crossing at different values of current/magnetic field. For the same reason, negative currents (i.e., a magnetic field directed against the propagation of light in the crystal) cause an opposite phase shift compared to that of positive currents, as shown in Figure 10. The zero crossing points of Δε_n_ for the negative currents are shifted relative to those for the positive currents, as labeled in Figure 10. Evidently, the Faraday rotation is added to the crystal intrinsic rotation for positive currents and subtracted for negative ones.

As mentioned, *Δε_n_* zero crossing points depend on the current/magnetic field which explains the slight discrepancy with the minima/maxima of the spectral polarimetric response, measured at *I_0_* = 0 A. Labels in Figure 10 show the average values of crossing points at ±(18, 20, 22, 24) A.

### 3.3. Temperature Stability

As shown in [11], the ellipticity and the zero crossings observed in our experiment depend on the birefringence and polarizer orientation. We found experimentally that the birefringence is mildly temperature dependent. This is probably due to a random distribution of mechanical stresses induced during the machining. In general, the temperature dependence of birefringence is minimal when the input polarization is parallel to the eigenaxis of the linear retarder. Having this in mind, we experimentally found an input polarization orientation where the spectral position of the zero crossing points does not change more than 0.3% over the temperature range −20 °C–+50 °C, as shown in Figure 11a. In that manner, they differ significantly from the minima/maxima of the polarimetric response (see Figure 5).

The zero crossing points can be considered as the operating points of the output SOP. Since they are independent of temperature, the phase shifts that result in the spectral changes between these two points are also independent of temperature. This was indeed observed experimentally and is illustrated in Figure 11b, showing the spectral dependence of the phase shift at currents of 24 A over the specified temperature range. The same behavior was observed for other current values.

The magnitude of the Faraday phase shift around 602 nm was 1.9 ± 0.1 deg for all temperatures, which agreed with the experimental precision of the temperature control ±1 °C.

### 3.4. Measurement Accuracy

It is necessary to comment on the accuracy of the measurements made by using channeled polarimetry. Channeled polarimetry goes some way towards overcoming temporal misregistration or intensity differences between successive measurements, which are a significant source of error. This detection method implements snapshot polarimetry [14] which allows the detector to measure the intensity at a time interval during which the phase of the complex amplitude is constant. However, it should be noted that the measurements are accompanied by errors inherent in the channeled polarimeter, with error sources specific to each of the Stokes parameters [19]. Due to the optimization of the retarder thicknesses, we were able to eliminate the crosstalk between channels, which is one of the error sources. In order to evaluate the influence of the other error sources, such as the alignment error of the retarders, we made successive independent measurements of the Stokes parameters at constant polarization states.

Precision polarization optics were utilized to conduct model experiments, with the objective of evaluating the accuracy of the results. The experimental set-up depicted in Figure 12 was employed to perform the measurements.

The model experiment consisted of ten sequential measurements of two linear SOPs with different polarization plane directions set by rotating a Thorlabs AHWP10M-600 achromatic half-wave plate to a fixed angle.

The standard deviation of all Stokes parameters was found to be almost the same. To determine the measurement accuracy of the phase shift induced by an optical rotator, the spectral dependence of the rotation angle of the polarization plane was calculated using the Formula (9), where atan2 is the four-quadrant inverse tangent.Θ = atan2(S_2_,S_1_)/2(9)

The results of the ten independent consecutive measurements (marked in different colors) are displayed in Figure 13.

The standard deviation up to 580 nm was 0.015 deg, and above that value it was estimated to be 0.01 deg at a fixed wavelength (e.g., 627.4 nm), which was consistent with the resolution of the spectrometer. It should be noted that the angle of rotation of the polarization plane varies over the spectral range. This is due to the imperfection of the achromatic half-wave plate and the inaccurate collimation of the light beam. This does not affect the evaluation of measurement accuracy, as it refers to a fixed wavelength in the spectral range considered.

The stated accuracy of the polarization plane rotation measurement is influenced by the accuracy of the S_1_ and S_2_ determinations. As the phase shift induced by the Faraday rotation is only determined by S_3_, it is safe to say that the measurement accuracy was higher than 0.01 deg.

## 4. Conclusions

To the best of our knowledge, channeled polarimetry has never been used before for data acquisition from magneto-optical current sensors. This method offers the unique opportunity to determine the operational point of the output polarization state, control its spectral position and temperature drift, and directly measure the phase shift induced by the Faraday effect. This was demonstrated in our experiment with a magneto-optical sensor based on a BSO crystal.

Magneto-optical sensors hold considerable promise for utilization as optical current transformers within high-voltage electrical power systems. In comparison to conventional current transformers, they exhibit notable advantages, particularly in the domains of measurement and protective relaying. Notwithstanding the intensive research conducted in this field, it remains challenging to fabricate an optical sensor that meets the requisite accuracy standards across a broad temperature range. The challenges associated with achieving this for magneto-optical sensors can be attributed to the following factors:-The optical properties of the sensing element are contingent on external conditions;-Detection of Faraday rotation by encoding it in intensity or spectrum does not allow the creation of a reference signal that effectively accounts for changes induced by external conditions.

Channeled polarimetry directly detects the spectral dependence of the phase shift in the absence of a magnetic field, thereby serving as a reference signal against which the real-time measurement is made. The experimental demonstration of temperature stability substantiates the efficacy of this approach.

The results reported herein should be interpreted as preliminary findings of an ongoing study. Primarily, the accuracy of the system currently achieved (0.01 deg) should be improved. For practical applications, it is necessary to increase the accuracy by one order of magnitude. To achieve this, data processing accuracy needs to be improved, i.e., the accuracy of the Fourier transform, the effective splitting of the channels at carrier frequencies, and the calibration procedure. This is currently being carried out and the results will be reported in our forthcoming article.

In addition, the issue of temperature stability was addressed in the present study. It was established that the influence of intrinsic optical activity and birefringence is negligible. However, further investigation is required to ascertain whether the method accounts for the temperature dependence of the Verdet coefficient. Given that its influence is estimated to be approximately 1.5% over a hundred-degree temperature range, it is imperative to enhance the accuracy of temperature control.

In summary, the efficient solutions offered by spectrally channeled polarimetry applied to optical current sensors would enable the widespread commercialization of these sensors.

## Figures and Tables

**Figure 1 sensors-25-00466-f001:**
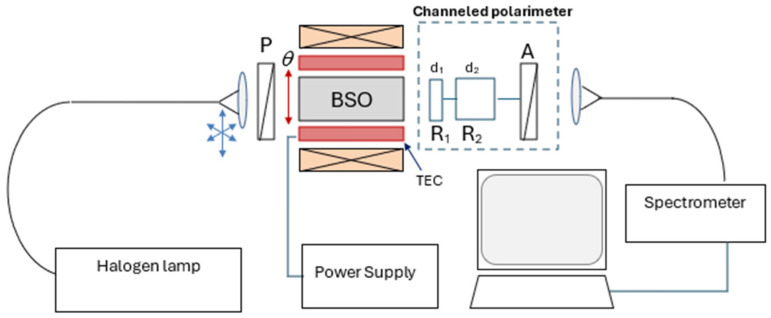
Experimental arrangement for channeled polarimetry: P—polarizer; θ—angle of orientation of polarizer; BSO—Bi_12_SiO_20_ crystal; TEC—thermoelectric coolers; R_1_ and R_2_—birefringent retarders; d_1_ and d_2_—thicknesses of retarders; A—analyzer.

**Figure 2 sensors-25-00466-f002:**
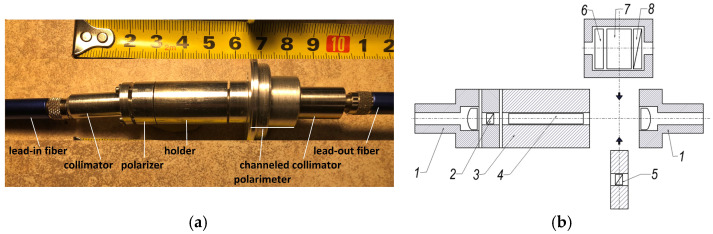
(**a**) Side view of the sensing element supplied with a channeled polarimeter. (**b**) Schematic drawing of the sensing element: 1—collimators; 2—polarizer; 3—holder; 4—BSO crystal; 5—analyzer; 6, 7—retarders with thickness d_1_ and d_2_, respectively; 8—analyzer; 6, 7, 8 make up the channeled polarimeter, which substitutes the analyzer for providing measurements in the present study.

**Figure 3 sensors-25-00466-f003:**
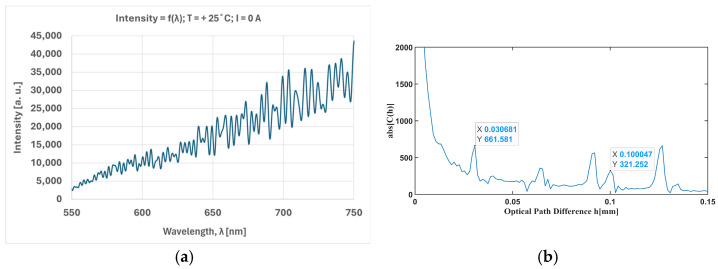
(**a**) Measured spectrum as modulated by the polarization interference. (**b**) Autocorrelation function.

**Figure 4 sensors-25-00466-f004:**
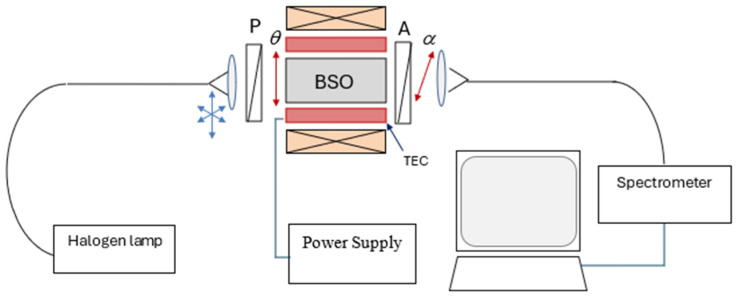
Experimental arrangement for spectrally interrogated polarimetry: P—polarizer; θ—angle of orientation of polarizer; BSO—Bi_12_SiO_20_ crystal; TEC—thermoelectric coolers; A—analyzer; α—angle of orientation of analyzer.

**Figure 5 sensors-25-00466-f005:**
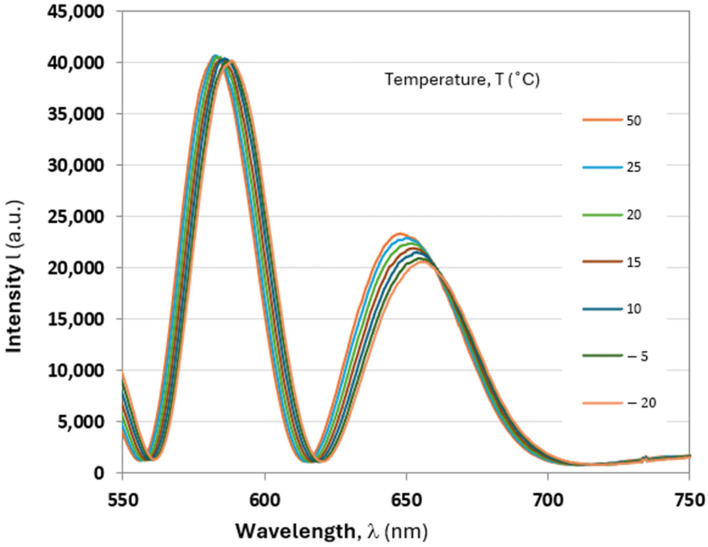
Response provided by spectrally interrogated polarimetry at *I* = 0 and temperature changes from −20 °C to +50 °C.

**Figure 6 sensors-25-00466-f006:**
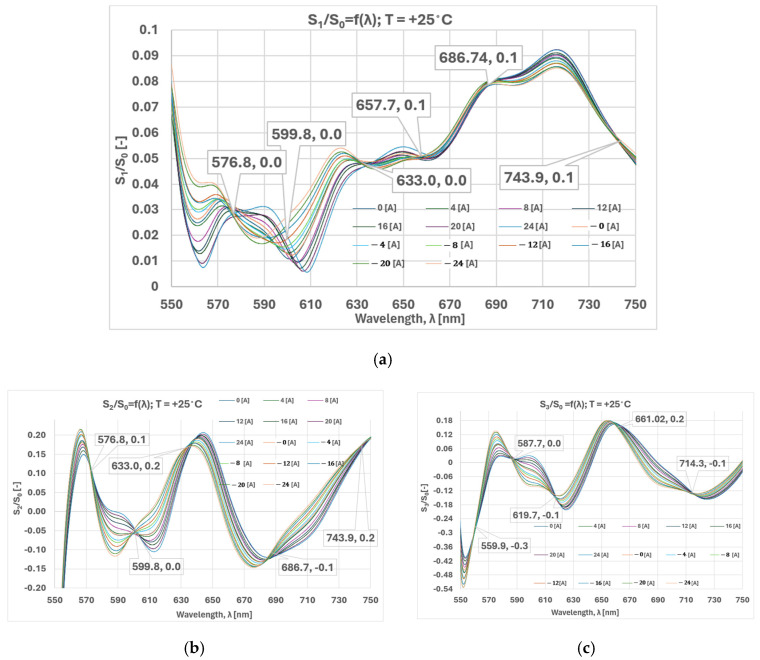
Spectral dependence of normalized Stokes parameters for current variation from −24 A to 24 A at 25 °C: (**a**) *S*_1_/*S*_0_ = *f*(*λ*); (**b**) *S*_2_/*S*_0_ = *f*(*λ*); (**c**) *S*_3_/*S*_0_ = *f*(*λ*).

**Figure 7 sensors-25-00466-f007:**
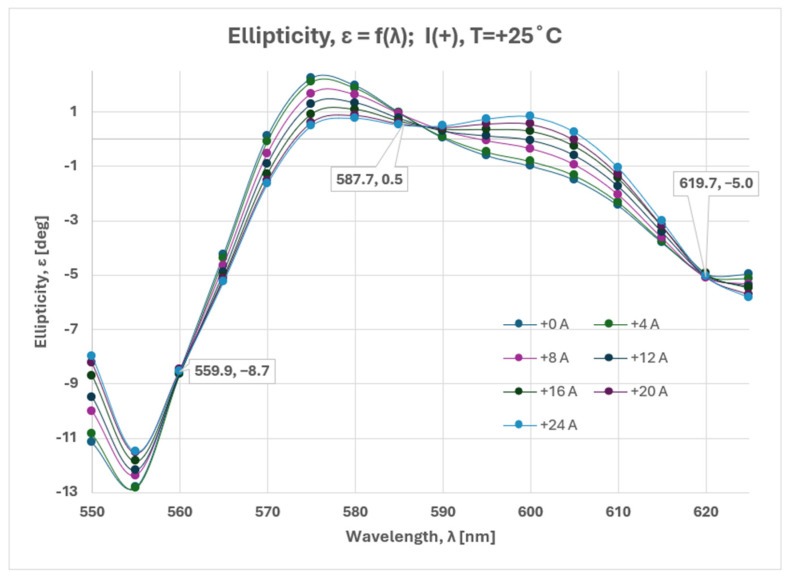
Calculated ellipticity for current variation from 0 A to 24 A at 25 °C.

**Figure 8 sensors-25-00466-f008:**
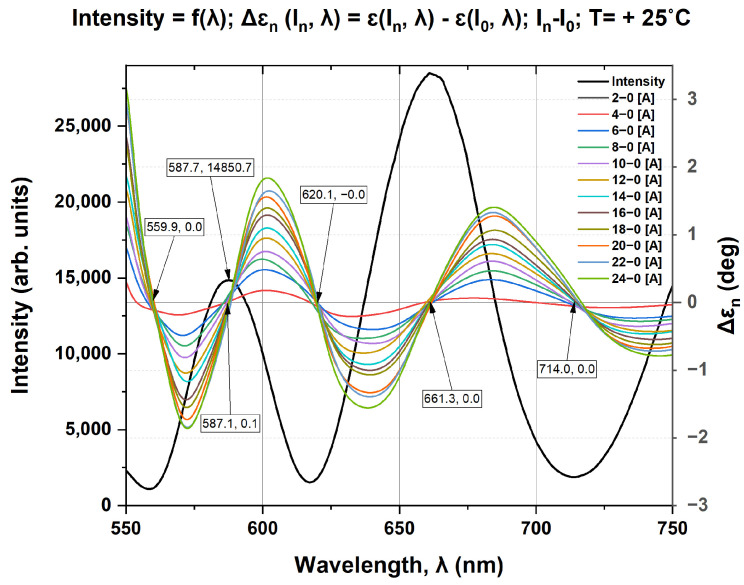
Correlation between Δε_n_ zero-crossing points (locus points of S_3_) for current variation from 0 A to 24 A and minima/maxima of the spectral polarimetric response at 25 °C.

**Figure 9 sensors-25-00466-f009:**
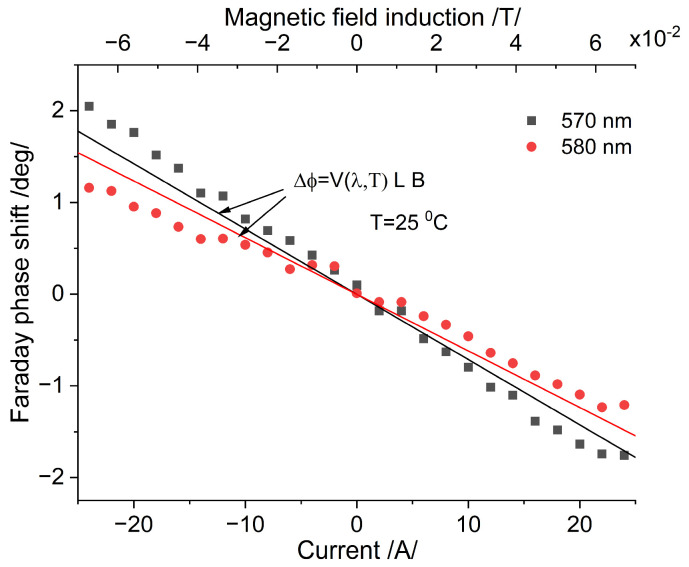
Faraday phase shift and experimental and theoretical data: dot curves and solid lines, respectively.

**Figure 10 sensors-25-00466-f010:**
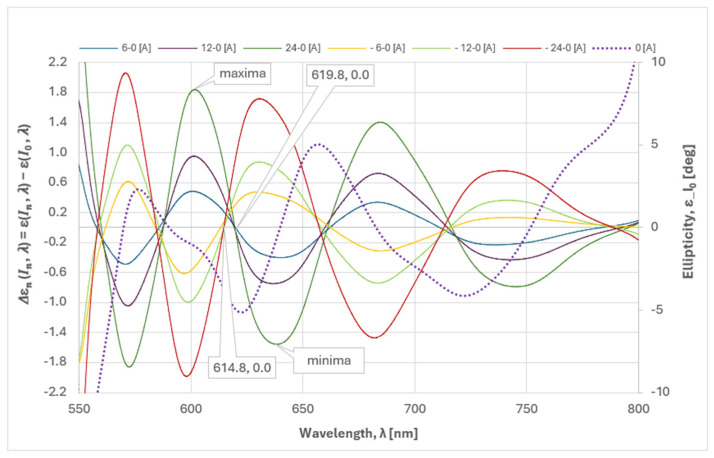
Effect of positive and negative currents on the sign of the Faraday phase shift and on the positions of *Δε_n_* zero-crossing points at 25 °C (dot curve represents ellipticity at I = 0 A).

**Figure 11 sensors-25-00466-f011:**
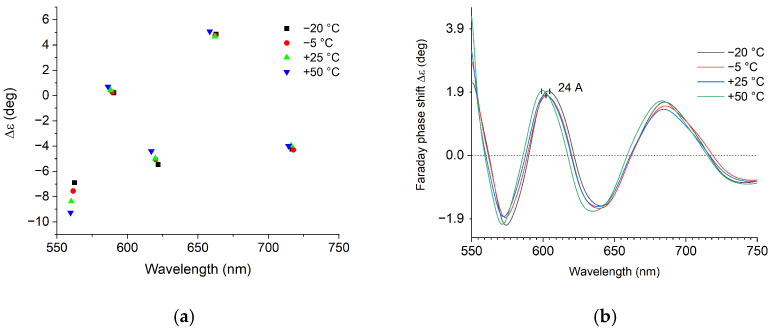
Experimentally observed temperature stability: (**a**) of the spectral position of the zero crossing points and (**b**) of the magnitude of the Faraday phase shift.

**Figure 12 sensors-25-00466-f012:**
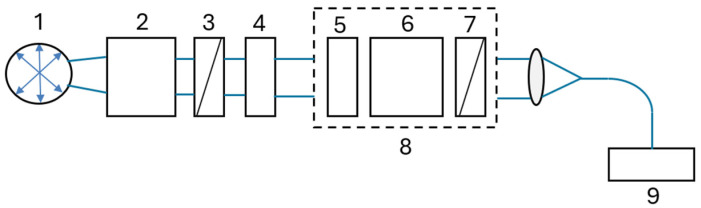
Set-up for measurement accuracy assessment experiment: 1—halogen lamp; 2—collimator; 3—polarizer; 4—achromatic half-wave plate; 5, 6—retarders, 7—polarizer; 8—channeled polarimeter; 9—spectrometer.

**Figure 13 sensors-25-00466-f013:**
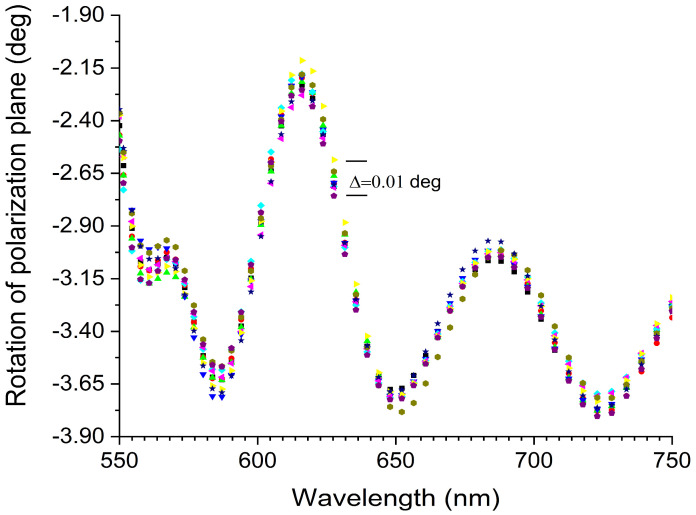
Results of the measurement accuracy assessment model experiment.

## Data Availability

Dataset available on request from the authors.

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
