# Peer review of "Channeled Polarimetry for Magnetic Field/Current Detection"

_sensors, 2025, doi:10.3390/s25020466_

Round 1

Reviewer 1 Report

Comments and Suggestions for Authors

The method of spectrally modulated polarimetry to investigate the magneto-optical response of the sensing element. It allows detection of the phase shift caused by Faraday rotation alone, making the detection independent of temperature.  I think it is very useful for current and magnetic field measurement in different situation. 

The experiment and results had better to be described more detail and clear.  

Reviewer 2 Report

Comments and Suggestions for Authors

Dear Author,

please, see the attached file with some minor comments.

Could you please specify some exact applications, features and benefits of your future current sensor? For example, use a simplified model of the nuclear power plant with the typical package of high-voltage - to - consumer transformer substations and cable lines.

Thank you in advance,

Reviewer 3 Report

Comments and Suggestions for Authors

The manuscript is devoted to the development of the magnetic field sensor based on the measurenet of polarization state of light. The high experimental accuracy and thermal stability of the sensor is shown. Nevertheless I have to point out many shortcomings:

1) It is not clear why the suggested type of the sensor is better than traditional devices for the magnetic field measurements, for example vibration magnetometer or Hall sensor.

2) The method of the Stokes parameter measurements is not well described in the manuscript. I guess that the Authors used the setup shown in Fig. 1. What are d_1 and d_2? How the analyzer was positioned? The procedure of the Stokes parameter measurements remains unclear. 

3) I would like to see in the manuscript more information about the BSO crystal. Does it has any spectral features in the region under study? The hysteresis loop of the crystal should be also presented here - it is crucially important for the magnetic field sensor.

4) Labels in the Figures 5 (a) and (b) are too small.

5) The authors use both "polariZation" and "polariSation"; "analyZer" and "analySer". 

6) Typos:

line 38: "Faradat"

Figure 2 caption: "bythe"

Figure 6, left axis and Figure 9, right axis: "Elipticity"

Line 311: "V.B"

7) About half of the References are to the authors' own works. I think the literature review is not comprehensive enough.

Reviewer 4 Report

Comments and Suggestions for Authors

The article entitled "Channeled Polarimetry for Magnetic Field/Current Detection" presents a novel approach to the polarimetric characterization of BSO using advanced spectral techniques. The study proposes two experimental setups for measuring polarization parameters under different optical conditions, with potential applications in optical sensors and birefringence studies induced by external fields. While the topic is relevant and has significant scientific potential, the manuscript exhibits notable issues in conceptual clarity, experimental justification, and graphical quality. These must be addressed to ensure the article meets the quality standards required for publication.

1.     The introduction predominantly references work conducted by the authors without situating the study within a broader research context. This self-contained approach may limit the perceived impact of the work.

2.     The difference between the experimental setups presented in Figures 1 and 3 is not adequately justified in the text. While it is mentioned that the first setup is used for spectrally modulated polarimetry and the second for spectrally interrogated polarimetry, the rationale for selecting one over the other, or whether the two setups are complementary or exclusive, remains unexplained.

3.     In Figure 3, the component labeled as "A" is neither identified nor explained in the text, leading to ambiguity regarding its function.

4.     Some figures lack clear explanatory links to the text. For instance, Figure 1 does not detail how the retarders influence spectral modulation, while Figure 6 fails to highlight critical features such as zero crossings. The connection between visual results and the conclusions presented needs to be strengthened to avoid potential misunderstandings.

5.     The retardance values of the retarders used in Figure 1 are not specified. Is this a critical omission since the retardance directly affects system sensitivity and could significantly impact the observed results?

6.     Conceptually incorrect statements about the effects of retarders on the circular components of polarization are present. For example, the manuscript claims that "The phase difference introduced by the retarders changes the circular components of the polarization." Retarders induce phase shifts between linear components, and changes in the circular components are an indirect consequence of transformations between bases. This fundamental error compromises the interpretation of experimental results and undermines the manuscript's scientific validity.

7.     Several sentences are confusing or technically ambiguous. For example:

"The phase difference introduced by the two elements in our sensor was not only dependent on their orientation, but also on the dispersion due to the polychromatic light source." It is unclear whether this dispersion dependence is purely linear or includes nonlinear effects.

"The ellipticity and the zero crossings observed in our experiment depend on the birefringence and polariser orientation." The interaction between polarizer orientation and other system elements to generate zero crossings is not clarified.

"The observed S3 ≠ 0 can only be explained by the presence of birefringence in the crystal due to residual stresses induced during machining and mounting." This explanation ignores potential contributions from optical alignment or external influences.

"The state of polarization was elliptical and the phase difference between the left and right polarization was introduced by both the optical linear retarder and the rotator." This implies the linear retarder directly affects circular components, which is incorrect and misleading.

8.     The parameter S1, which describes the difference between horizontal and vertical linear polarization components, is not calculated. This omission should be explicitly justified.

9.     Equation (1) does not specify whether the cosine function accounts for nonlinear or dispersive effects, which appear to be excluded. This could result in misunderstandings about the model's limitations, particularly when applied to more complex systems.

10.  Several figures have graphical quality issues. For example, Figure 2 features axis labels that are too small, and Figure 9 lacks annotations highlighting key features like maxima and minima, What program is used for graphing?

11.  The manuscript does not adequately justify the choice of circular over linear bases for describing polarization.

12.  The advantages of the proposed system over a commercial polarimeter are not discussed. Since commercial polarimetry solutions exist, the manuscript should highlight specific benefits of the proposed setup, such as improved sensitivity, reduced cost, or broader applicability.

13.  The characteristics of the light source are not described, and it is not discussed whether nonlinear polarization rotation (NPR) phenomena might influence the observed effects.

14.  The manuscript contains typographical errors, such as "spectometre" instead of "spectrometer" and "polarimetryc" instead of "polarimetric." Additionally, inconsistent spacing and punctuation appear in several sections.

The manuscript addresses a relevant topic but requires substantial revisions to address omissions, conceptual errors, and graphical issues before meeting the standards of a high-impact scientific publication.

Round 2

Reviewer 3 Report

Comments and Suggestions for Authors

The authors have done a good job on the text. The manuscript can be published in "Sensors".

Reviewer 4 Report

Comments and Suggestions for Authors

I would like to acknowledge the efforts made by the authors to address the comments raised during the initial review of the manuscript entitled "Channeled Polarimetry for Magnetic Field/Current Detection." Upon reviewing the revised version, I noted that the authors have implemented significant modifications that have notably strengthened the manuscript. In particular, I commend the inclusion of a new figure to clarify the experimental setups, as well as the detailed explanations provided regarding the key components of the system and their relevance to the results presented.

These improvements enhance the quality of the manuscript. However, I must highlight that some responses to the review comments were written in a tone that could be perceived as inappropriate or disparaging toward the reviewer. Such language detracts from the collaborative nature of the review process and risks diminishing the perceived value of what is otherwise a quality piece of work. I recommend that, in future communications, the authors maintain a respectful and professional tone, in alignment with the standards of academic courtesy.

Despite this issue, I consider the manuscript ready for publication in its current form.